# Point on the Aortic Bicuspid Valve

**DOI:** 10.3390/life12040518

**Published:** 2022-03-31

**Authors:** Chloé Bernard, Marie Catherine Morgant, David Guillier, Nicolas Cheynel, Olivier Bouchot

**Affiliations:** 1Department of Anatomy, University of Burgundy Medical School, 21000 Dijon, France; mariecatherine-morgant@chu-dijon.fr (M.C.M.); david.guillier@chu-dijon.fr (D.G.); nicolas.cheynel@chu-dijon.fr (N.C.); 2Department of Cardiovascular and Thoracic Surgery, Dijon University Hospital, 21000 Dijon, France; olivier.bouchot@chu-dijon.fr; 3Department of Digestive Surgery, Dijon University Hospital, 21000 Dijon, France

**Keywords:** bicuspid aortic valve, Siever’s classification, genetics, histology, aortic aneurism

## Abstract

Background—Bicuspid aortic valve (BAV) disease is the most prevalent congenital heart disease in the world. Knowledge about its subtypes origin, development, and evolution is poor despite the frequency and the potential gravity of this condition. Its prognosis mostly depends on the risk of aortic aneurysm development with an increased risk of aortic dissection. Aims—This review aims to describe this complex pathology in way to improve the bicuspid patients’ management. Study design—We reviewed the literature with MEDLINE and EMBASE databases using MeSH terms such as “bicuspid aortic valve”, “ascending aorta”, and “bicuspid classification”. Results—There are various classifications. They depend on the criteria chosen by the authors to differentiate subtypes. Those criteria can be the number and position of the raphes, the cusps, the commissures, or their arrangements regarding coronary ostia. Sievers’ classification is the reference. The phenotypic description of embryology revealed that all subtypes of BAV are the results of different embryological pathogenesis, and therefore, should be considered as distinct conditions. Their common development towards aortic dilatation is explained by the aortic media’s pathological histology with cystic medial necrosis. At the opposite, BAV seems to display a profound genetic heterogeneity with both sporadic and familial forms. BAV can be even isolated or combined with other congenital malformations. Conclusions—All those characteristics make this pathology a highly complex condition that needs further genetic, embryological, and hemodynamic explorations to complete its well described anatomy.

## 1. Introduction

Bicuspid aortic valve (BAV) is present in 1% of the general population. It is the most frequent cardiac congenital disease in the world [1]. Men are more affected than women (sex ratio 3:1) [2]. 

Despite this wide-spreading, their genetic [2], molecular, embryologic, and developmental origins remain unclear. 

The seriousness of this perplexing disease lies in the nine-fold higher risk of dissection or rupture secondary to aortic aneurysm development [3]. 

Aneurysm growth history seems to be multifactorial [4], and seems to associate both hemodynamic and histological origins, which reinforce the idea that BAV is a complex “bicuspid disease” that involve both aortic valve and aorta disorders [5]. Many BAV’s subtypes exist. They are defined by their morphology with different number of raphe and different commissure’s orientation. Every subtype appears to have a different embryological origin [6], evolution, and prognosis that results in different hemodynamic and functional evolution [7]. An emerging hypothesis is that every type of BAV leads to different aortic evolution and variable location of aortic aneurism, with different risk of dissection and rate of change. This aspect of the disease highlights the importance of defining all cases well to propose the best clinical management to BAV patients.

In this article, we aim to discuss the developmental mechanisms (embryology and genetics), histology, pathophysiology, classifications, and syndromic association of this disease with particular attention to the different existing subtypes.

This review should allow a better understanding of the “bicuspid disease” in way to further improve the everyday medical and surgical strategies; prevention, screening, surveillance, and aortic replacement depending on the subtype of BAV. In fact, this work aims to be a ramp for the study of clinical hypothesis on BAV anatomy, variation on leading aortic flow and histologic congenital or resultant modifications.

## 2. Methods

To collect the most significant amount of data about embryology, genetic, histology, and anatomy, we performed a systematic review of the literature with MEDLINE and EMBASE databases. MeSH terms used were “bicuspid aortic valve”, “aneurysm”, “ascending aorta”, “bicuspid aortic valve pathophysiology”, “bicuspid aortopathy”, “genetics”, “bicuspid histology”, “aortic syndrome” and “bicuspid classification”. 

Initially, we selected 52 studies about bicuspid aortic valves. In this case, 16 were excluded because their topic did not correspond to our expectations. No statistical analysis was achieved. 

## 3. Definition

The bicuspid aortic valve is an anatomical anomaly that associates a missing commissure and, consequently, an outnumbered leaflet. The number of leaflets depends on the commissure’s number. One or two commissures can be lacking. Sometimes, commissures are replaced by a raphe. Raphe is a kind of scar tissue or “fused area” between 2 underdeveloped cusps. We can enumerate five different kinds of BAV. They differ in commissure and raphe number, localization, and orientation. This complex process also involves the sinuses and the coronary orifice’s position.

In this article, we do not consider aortic valve with two raphes (one commissure) as a subtype of BAV. This valve configuration is assimilated into the unicuspid valve family.

It is essential to highlight that the valve is not the only aspect of the disease. Additionally, the aortic wall is abnormal and presents intrinsic histological anomalies that facilitate aneurism’s development. Aortic dilatation is found in 35–80% of cases [8,9], and preferentially concerned tubular aorta.

Due to both anomalies, two main complications make the future of BAV patients: valve degeneration with a high level of calcification and increased risk of aortic dissection secondary or not to dilatation. 

BAV represents a group of diseases that incorporate various phenotypes, etiologies, and pathogenesis that still require explorations for a deep understanding.

## 4. Description and Classifications

Till now, there was no official consensus on a phenotypic description, and many classifications have been proposed. Classifications differ with the description criteria that is taken into consideration; it can be the number of raphes and their position, the cusp position, the number of commissures, and their arrangement regarding coronary ostia. The reference for BAV classification was the Sievers one’s, published in 2007 [10]. The authors described the valve by taking into consideration the number of raphes and their orientations, as we can see in Figure 1. In their article, Sievers et al. defined BAV as a deformed aortic valve with less than three zones of parallel apposition between the cusps. This definition includes two categories of BAV; “true” BAV made of two cusps and no raphe and “false” BAV made of three cusps and one or two raphes. 

Type 0 denotes ‘true’ BAVs, whereas type 1 (1 raphe) and type 2 (2 raphes) denote ‘false’ BAVs.

Recently, an international classification on BAV has been proposed to standardize the description and identified 3 types of BAV. It described a fused type (with right-left cusp fusion, right-non coronary fusion and left-non coronary fusion), the two sinus type (latero-lateral and antero-posterior) and the partial-fusion type [11]. 

Other authors, such as Brandenburg et al., divided different types of BAV on commissures’ locations and then, on raphes’ positions [12]. Sabet et al. only take into consideration raphes’ situations [13], whereas Roberts et al. [14], and Kang et al. [15] used cusp and raphes’ positions to define their BAV subtypes.

For a minor part of teams, the valve cannot be analysed individually, and the morphology of the root and ascending aorta needs to be considered. Schaefer et al. associate the description of valve anatomy with three kinds of aortic shapes; «normal» called type “N” (diameter of the sinus of Valsalva (SV) is greater than the diameter at the sinotubular junction and greater than or equal to the width of the ascending aorta), “ascending distension” called type “A” (SV diameter > junction diameter, but less than ascending one) and “effaced” morphology called type “E” (SV diameter ≤ junction diameter) [16]. Then, they mixed both valve and aortic characteristics to create nine subgroups of BAV morphologies. In their series, type A with right-left fusion was predominant. Those different aortic shapes can be the result of different development and tissue composition or maybe the results of different aortic blood flow. Indeed, valve-opening orientation in BAV leads to different corresponding aortic jet shapes.

Concerning the influence of leaflet geometry on the risk of aneurysms development, opinions differ. For Boodhwani et al., the type 0 of Sievers compared with the type 1 was associated with less wall shear stress both in the proximal (*p* = 0.009) and mid-ascending aorta (*p* = 0.027) [16]. Those differences possibly influence aneurism’s development.

Even if dilatation can occur in every aortic section, the proximal ascending aorta is the most frequently affected site of dilation [17]. 

We can speculate that the risk of dissection or aortic dilatation depend on the BAV morphology, but this remains to be shown. For Jackson et al., and many other teams, aortic morphology is not influenced by BAV type [18,19].

In this article, we decided to realize a summarize of every subtype of BAV and to describe their frequency and specificity, particularly associated risk, dysfunction, or other disorders (Figure 2). It is important to note that there is a spectrum of missing, underdeveloped, or developed commissures and adjacent cusps [10] that cannot be considered.

BAV can be divided in two main subtypes of valves: true and false BAV. True BAV represents 7 to 11% of BAV [19,20] and included two subtypes depending on the commissure’s orientation. There, the aortic valve is composed of 2 cusps separated by one major commissure with no raphe. In the anterior-posterior type (AP), both coronary ostia are located above the same cusp, whereas in the left-right type (LR), each coronary ostia are situated above its cusp.

Repartition of those two subtypes is heterogeneous between articles, but every article agrees that LR type is more frequent (4 to 10%) [20,21] than AP type (0.5 to 3.3%) [20,21].

False BAV represents up to 88% of BAV [20]. This category is divided into three subtypes of valves that always had two commissures and one raphe. Raphe’s location differentiates them. The left-right (LR) type corresponds to the fusion of both cusps that are located below both coronary ostia. In this case, two commissures exist; one separates the non-coronary (NC) cusp from the free edge next to the left ostia and another one separates the NC cusp from the free side next to the right ostia. This subtype is the most frequent of false BAV; 68–80% [22,23]. It is the most studied type in the literature, and authors agree to say that it is linked with a rapid progression of aortic dilatation compared with other BAV [17,24]. Moreover, it appears that this conformation is favourable to earlier valve dysfunction with significant valve regurgitation [9,25]. Opinions differ concerning the predominant location of aortic dilation; for Verma et al., the tubular ascending aorta is the privileged target of aneurysms [26], whereas it is the annulus and the sinus of Valsalva for Schaefer et al. [3].

On a more general level, for Kong et al. [23], the degree of aortic regurgitation and dysfunction is directly due to the raphe’s number. They explained that in the absence of raphe, the aortic valve remains functional for a longer time than if there is one or, even worse, two raphes.

For teams which consider valves with only one commissure as a bicuspid valve, only one configuration exists and it represents 0.6 to 5% of BAV cases [20,22]. This anomaly is highly pathologic and requires earlier surgery compared to other BAV types [7].

## 5. Embryology

Some morphogenetic studies showed that different subtypes of BAV are developed at different embryonic steps. So, they can be viewed as etiologically distinct entities. For example, Anderson et al. demonstrated that RN and LR BAVs have different pathogenesis in a murine model [27].

Aortic valve development is a complex process that involves the growth and differentiation of endocardial cells and their migration after epithelial-to-mesenchymal transformation to form endocardial cushions at the atrioventricular canal and within the outflow tract. Neural crest and secondary heart field-derived cells participate in valve development, too, particularly in the formation of the outflow tract (OT) [28]. 

The healthy aortic valve development begins with the outflow tract septation, which represents both the future aorta and the pulmonary artery. This separation results from the fusion of conotruncal bulges that grow inside the efferent tube for its entire height and lead to septum inception. Semilunar valves come from 2 endocardial cushions that developed in opposite quadrants of the efferent tract; septal (conotruncal cushions) and parietal (intercalated cushions) OT ridges (Figure 3). In the distal part of the conus, those cushions excavate themself to create semilunar sinus drafts at the future aorta and pulmonary artery’s origins. Similar cavities grew symmetrically on the aortic and pulmonary side of the conotruncal septum. Those cavities and the tissues between them will form both aortic and pulmonary valves. Finally, cuspids derive from endocardial cushions, but we know that interaction between neural crest cells and endocardium is necessary. 

Concerning the BAV genesis, observations of mouse embryos indicate that RN BAV is the result of a defective formation of the embryonic outflow tract endocardial cushions. In pathological embryology, before the OT septation, the posterior margin of the septal ridge and the posterior intercalated cushion fused and developed as a unique cushion and will lead to a unique cusp development instead of two. This abnormal unique cushion will become an abnormal RN aortic valvular leaflet. The remaining leaflet will become a regular left coronary cusp [29].

Hamster embryos’ observation denotes that LR BAV results from a deficient OT septation. The septal and parietal OT ridges and the posterior intercalated cushion show a usual arrangement before the septation. Then, the posterior margins of the ridges become extra fused, so a single anterior cushion forms, and will give the LR fused aortic valve leaflet. The non-coronary leaflet derives from the normal posterior intercalated cushion. 

Groenendijk et al. highlight that during normal cardiogenesis, endothelium-derived nitric oxide synthetase (eNOS) expression is restricted to endocardial cells and is shear stress-dependent [30]. Endothelium-derived nitric oxide is the mediator of endothelial cell podokinesis [31]; an essential step of the cardiac jelly’s colonization by endocardial cells to form the endocardial cushions. This step appears to be shear stress-dependent too. Knowing this, Fernández et al. hypothesized that eNOS deficiency in mice might alter endocardial cell migration and lead to an abnormal development of the valve cushions [29]. Authors suggested that an unusual behaviour of cardiac neural crest cells might be implicated in this event, such as Siu et al. [28], but the mechanism remains unclear. 

So, RN BAV appears caused by a pathologic OT cushions development, probably due to an increased nitric oxide-dependent transformation whereas, LR BAV seems to be the result of an anomalous embryonic OT septation, probably produced by neural crest cell behaviour’s alterations. However, we must keep in mind that those conclusions are based on murine analyses. In recent studies, it appears that genetic factors could be the leading cause of maldevelopment [32]. 

## 6. Histology

In BAV, the tendency to aortic dilatation can be explained by histological abnormalities, mainly situated in the aortic media. Classical abnormalities included; severe cystic medial necrosis, elastic fragmentation, leading to a thinner elastic lamellae and smooth muscle cells (SMC) reorientation [33]. Bauer et al. additionally showed that BAV patients had a greater distance between elastic lamellae [34]. All those anomalies can be attributed to a decrease in Fibrillin 1 and Matrix metalloproteinase 2 (MPP2) increased activity in BAV aortic media [33]. MPP2 controls the degradation of the extracellular matrix (ECM), which plays a major role to maintain the structural integrity of the vascular wall [6]. It is a cascading effect, because abnormal processing of Fibrillin 1 by vascular SMC initiates their own detachment from the ECM, lead to the release of MMPs together with their tissue inhibitors (TIMPs). The resulting matrix disruption, elastin, and lamellar fragmentation lead to increased apoptosis of vascular SMC and separation of the media layer, adversely affecting the structural integrity and flexibility of the aorta [25]. It appears that TIMPs’ expression controls the ECM metabolism. So, the balance between MMPs and TIMPs regulates ECM degradation in normal and pathological states and can lead to aneurysm formation when the balance is tipped towards increased MMP activity [35]. Authors agree that MMPs activity is increased in BAV patients’ aortic wall compared with healthy subjects, while TIMP-1 is reduced [36].

The remaining issue is to understand if this imbalance is genetically programmed or not. A hypothesis could be that BAV leads to abnormal aortic flow, which involves wall shear stress that created conditions suitable for increasing MMP activity that sustained a pathological circle leading to aortic dilatation. Yassine et al. observed histologic diversity in different segments of the enlarged ascending aorta of the same patient. The outer curve of the aorta is more frequently dilated in BAV patients. This condition is often explained by more intense shear stress due to an asymmetric flow across the valve. Moreover, this stress appears to exhibit greater elastic fibre fragmentation, to reduce collagen types I and III expression, and SMC apoptosis. MMPs and TIMPs are also variously expressed in all the BAV aortic wall; higher levels are seen in the concavity of the ascending aorta. These variations are also seen in microRNA expression. These findings strongly suggests that active cellular processes are involved in the development of bicuspid aortopathy, perhaps or even probably, mediated by hemodynamic forces [37].

## 7. BAV Genesis: Genetic, Associated Anatomic Anomalies, and Syndromes

BAV is a complex pathology with possible sporadic and familial transmission. Its expression can be variable and can appear as an isolated defect or associated with other congenital defects. Numerous genes mutations can lead to BAV, which explains both its high prevalence in the general population and their numerous shapes. Those mutations are usually unspecific and can induce other cardiac defects or syndromes, as represented in Table 1.

For example, mutations of NOTCH1 (9q34.3) are implicated both in familial BAV and in approximately 4% of sporadic cases [24]. This NOTCH1 insufficiency has a significant role during embryonic development of cardiac valves. Its dysregulation could predispose to other congenital cardiac malformation in particular the outflow tract [38].

BAV mutations can occur de novo, leading to sporadic forms or can be transmitted with an automosal mechanism with reduced penetrance [39], leading to familial clustering [24]. Historically, authors described 10–30% of familial form in BAV cases [39], but more recent studies established the heritability of BAV up to 89% [40,41,42]. Despite the well-established heritability of BAV, no single gene model exists that could explain the inheritance of this cardiac malformation. In fact, many discrete genes with various inheritance patterns probably also act in synergy as a polygenic trait [43]. It is may be the case for the aortopathy associated with BAV [6]. Giusti et al. identified, for the first time, a specific FBN1 mutation in BAV patients linked with aortic dilation (Marfan syndrome and other more severe fibrillinopathies were clinically excluded). FBN1 encodes a glycoprotein of the ECM involved in maintaining elastic fibres and in the anchorage of epithelial cells to the interstitial matrix. A decreased in FBN1 mRNA, or protein content, has been demonstrated in a subgroup of BAV, suggesting this gene to be one of those possibly associated with BAV [24].

ACTA2 gene mutation, a gene encoding smooth muscle a-actin, and known to be related to familial TAA, have also been detected in patients with BAV [44].

Many syndromes are associated with a high BAV prevalence; Turner, Williams (5–12%), Andersen [24], Bosley-Salih-Alorainy [24], and Athabascan Brainstem Dysgenesis syndromes’ [24]. Almost 50% of aortic coarctation are associated with a BAV, such as ventricular septal defect, patent ductus arteriosus, and aortic arch interruption even less so. Connective tissue disorders are also included; familial thoracic aortic aneurysms (TAA), Marfan Syndrom, Loeys-Diets Syndrom, Ehlers-Danlos Syndrom [24].

It appears that patients with LR BAV are more likely to present a severe aortic wall degeneration than patients with RN BAV. BAVs, excepted RN ones, are significantly associated with aortic coarctation, probably due to cardiac NC anomaly. 

Finally, BAV seems to display a profound genetic heterogeneity, which suggests the role of various discrete genes in its genesis. Animal studies have shown evidence of a combination of many genetic variants acting as a burden and epigenetic and environmental factors. This complex set of genetic and non-genetic factors may be likely to cause this extremely variable phenotypic expression of BAV [24].

Family-based genome linkage analyses were unable to identify a specific gene responsible for BAV. The issue is that in none of these studies, RN and LR BAVs were regarded as different conditions. 

## 8. Conclusions

The BAV is a pathology which becomes more and more interested in cardiology and in cardiac surgery because of its high incidence and its various consequences; aneurisms and valvular stenosis leading to surgical management.

The bicuspid aortic valve is a complex condition with many subtypes and many ways to be described. Sievers classification stays the gold standard. The literature agrees that the LR type is the commonest subtype of BAV. This review highlights a possible multigenic origin of this pathology with a potential role of epigenetic. Concerning aortic aneurysms development, the link is well established, but questions remain about factors that promote the growth, the localization, and the risk of rupture of those aneurysms. It appears major to understand if histology, wall shear stress, or hemodynamic abnormalities are responsible for or not only. A self-perpetuating mechanism is a hypothesis in this condition. Further studies are required to clarify all those remaining issues to provide clear BAV patient’s management. 

## Figures and Tables

**Figure 1 life-12-00518-f001:**
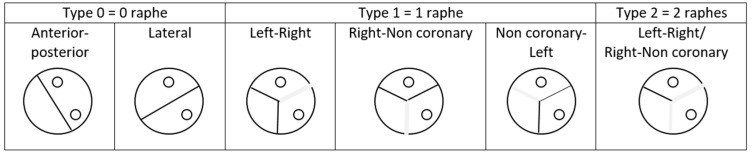
BAV classification, according to Sievers.

**Figure 2 life-12-00518-f002:**
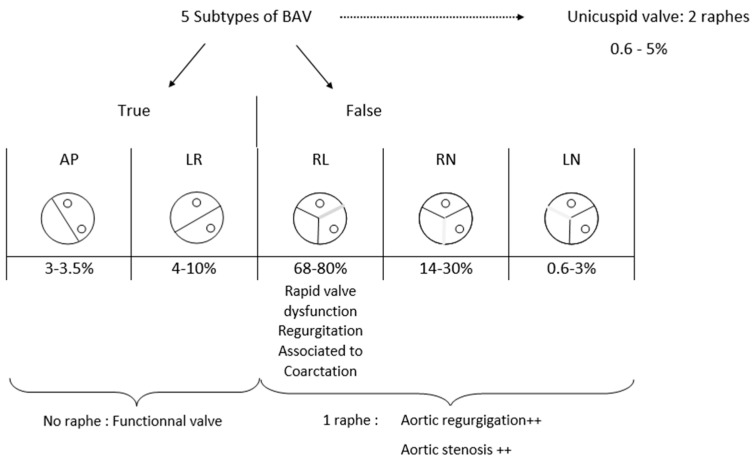
Bicuspid aortic valve subtypes anatomy and distribution.

**Figure 3 life-12-00518-f003:**
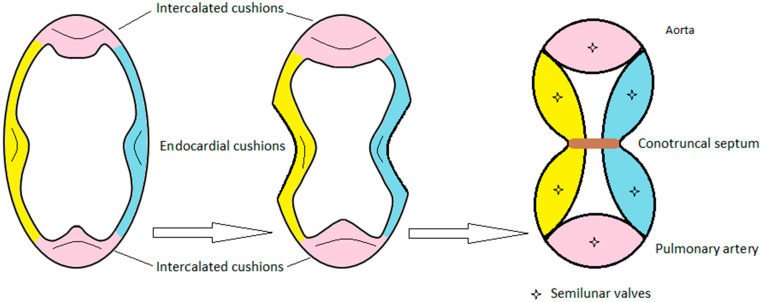
Embryology of the aortic valve.

**Table 1 life-12-00518-t001:** Genes and Syndromes associated with BAV.

GENES	CARDIAC DEFECT/ASSOCIATED SYNDROM
5q, 9q34, 13q, 15q25, 1-26, 18q	BAV
GATA5/GATA4	BAV
NOS3	BAV
PDIA2	BAV
AXIN1	BAV
NKX2.5	BAV
EGFR	BAV
ENG	BAV
TEX26	BAV
SMAD6	BAV
NOTCH1	BAV + Severe aortic calcifications
FGF8	BAV + Coronary, aortic/pulmonary artery
UFD1L	BAV + Aortic aneurisms
HOXA1	BAV + Bosley-Salih-Alorainy syndrome Athabaskan brainstem dysgenesis syndrome
FNB1	BAV + Marfan syndrome
ELN	BAV + Cutis laxa
ACTA2	BAV + Familial TAA
TGFb1/TGFb2	Loeys-Dietz syndrom + sporadic BAV
FLNB	BAV + Larsen syndrom
KMT2D, KDM6A	BAV + Kabuki syndrom
KCNJ2	BAV + Andersen-Tawil
22q11.2 deletion	BAV + DiGeorge
45 X0 karyotype	BAV + Turner syndrome
COL3A1	BAV + Vascular Ehlers Danlos syndrome
LIP2, ELN, GTF2I, GTF2IRD1, and LIMK1	BAV + William Beuren syndrome

BAV, Bicuspid aortic valve; TAA, Thoracic aortic aneurism.

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
