# Peer review of "Point on the Aortic Bicuspid Valve"

_life, 2022, doi:10.3390/life12040518_

Round 1

Reviewer 1 Report

The review titled “Point on the Aortic Bicuspid Valve: a Systematic review”, described the various hypothesis on the developmental mechanisms of the different subtypes of bicuspid aortic valve disease.

The review is interesting and well done even if not many original and innovative.

Author Response

Dear reviewer, thanks for your very constructive comments. 

This review was done by a surgeon for surgeons. It appears to us important to well understand this disease because multiplenhypothesis are developping about the various bicuspid valves anatomy and the aortic aneuvrysm developpement. For example, is the geometry of the valve the first step leading to different site of aortic aneuvrysm developpement due to modification in aortic flow? 

But, the anatomy doesn't seem to be the only factor and further investigations in histologic field are required. Currently, we don't know if the histology in initially different or if the modification of the aortic flow lead to histologic modifications. 

This work aims to be a ramp for further explorations in the surgical field. 

Reviewer 2 Report

The authors reported a review regarding aortic bicuspid valve. This manuscript is well written as a textbook.

Although the aims of this manuscript were to describe this complex pathology in way to improve the bicuspid patients’ management, they only concluded that further studies are required to clarify all those remaining issues to provide clear BAV patients’ management.

As a cardiac surgeon, it might be important to know that LR type is the most frequent of false BAV, and in linked with a rapid progression of aortic dilatation compared with other BAV.

I wonder what kind of new information the readers can get from this study specially in the clinical managements because all of the reports in the references are well known.

Author Response

Dear reviewer, thanks for your very constructive comments. 

This review was done by a surgeon for surgeons. It appears to us important to well understand this disease because multiple hypothesis are developping about the various bicuspid valves anatomy and the aortic aneuvrysm developpement. For example, is the geometry of the valve the first step leading to different site of aortic aneuvrysm developpement due to modification in aortic flow? 
But, the anatomy doesn't seem to be the only factor and further investigations in histologic field are required. Currently, we don't know if the histology in initially different or if the modification of the aortic flow lead to histologic modifications. 
This work aims to be a ramp for further explorations in the surgical field. 

More over, thanks you for your comment about LR type and its frequency and clinical consequences. We hoped that the present sentences were clear but probably not. 
" Repartition of those two subtypes is heterogeneous between articles, but every article agrees that LR type is more frequent (4 to 10%) [22] [23] than AP type (0.5 to 3.3%) [22] [23]." (Line 187-188). "It appears that patients with LR BAV are more likely to present a severe aortic wall degeneration than patients with RN BAV." (Line 348-349) 

Finally, we hoped that this article was both exhaustive for anatomic field and targeted concerning histology and genetic because this article is destinated to surgeon who are usually not specialist of those 2 fields. 

Reviewer 3 Report

  • In the review article, Bernard and his co-authors provide a detailed overview of the embryological and histological aspects of the bicuspid aortic valve.
  • They also describe the different classifications of BAV, but do not mention the clinical implications of the different classifications and the subtypes, especially when it comes to valvular stenosis requiring treatment.
  • Unfortunately, the manuscript is very hard to read. I am not a native English speaker either, but I strongly advise you to send in your manuscript for proofreading before re-submission. The manuscript cannot be accepted in the current version.
  • Also, the images were provided in very low quality. I hope this is a conversion issue, otherwise you will need to rework the images and make them at least 300 dpi.
  • Please change the title to Point on The Bicupsid Aortic Valve...

Author Response

Dear reviewer, thanks for your very constructive comments. 

This review was done by a surgeon for surgeons. It appears to us important to well understand this disease because multiple hypothesis are developping about the various bicuspid valves anatomy and the aortic aneuvrysm developpement. For example, is the geometry of the valve the first step leading to different site of aortic aneuvrysm developpement due to modification in aortic flow? 

But, the anatomy doesn't seem to be the only factor and further investigations in histologic field are required. Currently, we don't know if the histology in initially different or if the modification of the aortic flow lead to histologic modifications. 

This work aims to be a ramp for further explorations in the surgical field. 

Concerning subtypes and clinical consequences, we hopes that we answered in the 4th paragraph with the BAV classifications. Today, the surgical management doesn't depend on the type of BAV subtypes and there are only hypothesis about the impact of different subtypes on clinic, there is no clinical proof. 

Thanks a lot for your remarks about the english language; in that respect a native english (UK) red and corrected our article. 

We modified the title and our pictures on your advice.

Round 2

Reviewer 3 Report

Thank you for re-submitting your revised manuscript. I still think English proofreading would enrich the manuscript. Otherwise it can be accepted.

Congratulations!